# A massive quiescent galaxy at redshift 4.658

Adam C. Carnall[1✉], Ross J. McLure[1], James S. Dunlop[1], Derek J. McLeod[1], Vivienne Wild[2], Fergus Cullen[1], Dan Magee[3], Ryan Begley[1], Andrea Cimatti[4,5], Callum T. Donnan[1], Massissilia L. Hamadouche[1], Sophie M. Jewell[1] & Sam Walker[1]

The extremely rapid assembly of the earliest galaxies during the first billion years of cosmic history is a major challenge for our understanding of galaxy formation physics[1–5]. The advent of the James Webb Space Telescope (JWST) has exacerbated this issue by confirming the existence of galaxies in substantial numbers as early as the first few hundred million years[6–8]. Perhaps even more surprisingly, in some galaxies, this initial highly efficient star formation rapidly shuts down, or quenches, giving rise to massive quiescent galaxies as little as 1.5 billion years after the Big Bang[9,10]. However, due to their faintness and red colour, it has proven extremely challenging to learn about these extreme quiescent galaxies, or to confirm whether any existed at earlier times. Here we report the spectroscopic confirmation of a massive quiescent galaxy, GS-9209, at redshift, $z = 4.658$, just 1.25 billion years after the Big Bang, using the JWST Near-Infrared Spectrograph (NIRSpec). From these data we infer a stellar mass of $M_* = 3.8 \pm 0.2 \times 10^{10} \, M_\odot$, which formed over a roughly 200 Myr period before this galaxy quenched its star-formation activity at $z = 6.5^{+0.2}_{-0.5}$, when the Universe was approximately 800 Myr old. This galaxy is both a likely descendent of the highest-redshift submillimetre galaxies and quasars, and a likely progenitor for the dense, ancient cores of the most massive local galaxies.

During the past 5 years, several studies have identified GS-9209 as a candidate high-redshift massive quiescent galaxy[11,12], on the basis of its blue colours at wavelengths $\lambda = 2$–8 μm and non-detection at millimetre wavelengths[13]. GS-9209 is also not detected in X-rays[14], at radio wavelengths[15] or at $\lambda = 24$ μm (ref. 16). The faint, red nature of the source (with $H$ and $K$-band apparent magnitudes of $H_{AB} = 24.7$ and $K_{AB} = 23.6$) means that near-infrared spectroscopy with ground-based instrumentation is prohibitively expensive. The James Webb Space Telescope (JWST) Near-Infrared Spectrograph (NIRSpec) data, shown in Fig. 1, reveal a full suite of extremely deep Balmer absorption features, with a Hδ equivalent width, as measured by the Hδ$_A$ Lick index, of 7.9 ± 0.3 Å, comparable to the most extreme values observed in the local Universe[17]. These spectral features strongly indicate that this galaxy has undergone a sharp decline in star-formation rate (SFR) during the preceding few hundred million years.

The spectrum exhibits only the merest suspicion of [O II] 3,727 Å and [O III] 4,959 Å, 5,007 Å emission, and no apparent infilling of Hβ or any of the higher-order Balmer absorption lines. However, as can be seen in Fig. 2, both Hα and [N II] 6,584 Å are clearly, albeit weakly, detected in emission, with Hα also exhibiting an obvious broad component. This broad component, along with the relative strength of [N II] compared with the narrow Hα line, indicates the presence of an accreting supermassive black hole: an active galactic nucleus (AGN). However, the extreme equivalent widths of the observed Balmer absorption features indicate that the continuum emission must be strongly dominated by the stellar component.

To measure the stellar population properties of GS-9209, we performed full spectrophotometric fitting using the Bayesian Analysis of Galaxies for Physical Inference and Parameter EStimation (Bagpipes) code[18] (Methods). Briefly, we first masked the wavelengths corresponding to [O II], [O III], narrow Hα and [N II], due to likely AGN contributions. We then fitted a 22-parameter model for the stellar, dust, nebular and AGN components, as well as spectrophotometric calibration, to the spectroscopic data in combination with multiwavelength photometry. Throughout the paper we report only statistical uncertainties on fitted parameters. It should be noted however that systematic uncertainties in galaxy spectral energy distribution analyses can be substantially larger[19–21]. We investigate the effect of our choice of star-formation history (SFH) model in the Methods section.

The resulting posterior median model is shown in black in Figs. 1 and 2. We obtained a stellar mass of $\log_{10}(M_*/M_\odot) = 10.58 \pm 0.02$, under the assumption of a Kroupa initial mass function (IMF)[22]. We also recovered a very low level of dust attenuation, with a $V$-band attenuation in magnitudes of $A_V = 0.02 \pm 0.02$. The SFR we measured averaged over the past 100 Myr is consistent with zero, with a very stringent upper bound, although this is largely a result of our chosen SFH parameterization[19]. We provide a detailed discussion of the SFR of GS-9209 in Methods.

The SFH we recovered is shown in Fig. 3. We found that GS-9209 formed its stellar population largely during an approximately 200 Myr period, from around 600–800 Myr after the Big Bang ($z \approx 7$–8). We recovered a mass-weighted mean formation time, $t_{form} = 0.76 \pm 0.03$ Gyr after the Big Bang, corresponding to a formation redshift of $z_{form} = 6.9 \pm 0.2$. This is the redshift at which GS-9209 would have had half its current stellar mass, approximately $\log_{10}(M_*/M_\odot) = 10.3$. We find that GS-9209 quenched (which we define as the time at which its specific

[1]Institute for Astronomy, School of Physics & Astronomy, University of Edinburgh, Royal Observatory, Edinburgh, UK. [2]School of Physics & Astronomy, University of St Andrews, St Andrews, UK. [3]Department of Astronomy and Astrophysics, UCO/Lick Observatory, University of California, Santa Cruz, USA. [4]Department of Physics and Astronomy (DIFA), University of Bologna, Bologna, Italy. [5]INAF, Osservatorio di Astrofisica e Scienza dello Spazio, Bologna, Italy. ✉e-mail: adam.carnall@ed.ac.uk

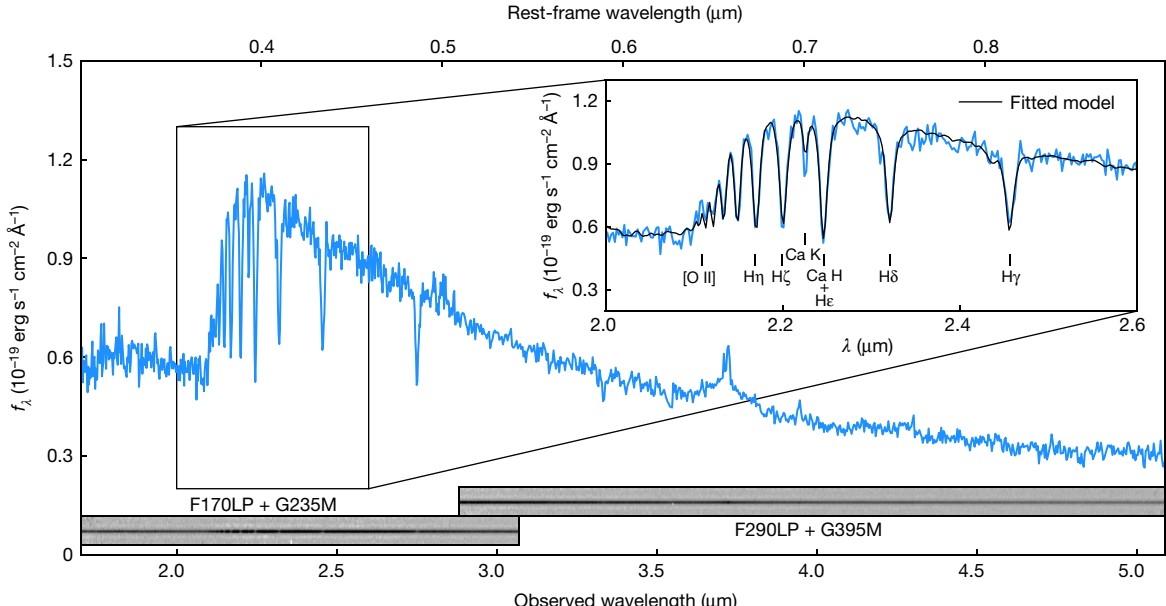

**Fig. 1 | JWST NIRSpec observations of GS-9209.** The figure shows flux per unit wavelength ($f_\lambda$) as a function of wavelength, $\lambda$. Data were taken on 16 November 2022, using the G235M and G395M gratings ($R = 1,000$) with integration times of 3 hours and 2 hours, respectively, providing wavelength coverage from $\lambda = 1.7$–$5.1\,\mu m$. The galaxy is at a redshift of $z = 4.6582 \pm 0.0002$, and exhibits extremely deep Balmer absorption lines. The spectrum strongly resembles that of an A-type star and is reminiscent of lower-redshift post-starburst galaxies[40–42], clearly indicating that this galaxy experienced a substantial, rapid drop in SFR in the past few hundred million years. The spectral region from $\lambda = 2.6$–$4.0\,\mu m$, containing H$\beta$ and H$\alpha$, is shown at a larger scale in Fig. 2.

star-formation rate (sSFR) fell below 0.2 divided by the Hubble time[23] at time $t_{quench} = 0.83^{+0.08}_{-0.06}$ Gyr after the Big Bang, corresponding to a quenching redshift of $z_{quench} = 6.5^{+0.2}_{-0.5}$.

Our model predicts that the peak historical SFR for GS-9209 (at approximately $z_{form}$) was within the range $SFR_{peak} = 490^{+680}_{-300}\,M_\odot\,yr^{-1}$. This is similar to the SFRs of bright submillimetre galaxies (SMGs). The number density of SMGs with a SFR of more than $300\,M_\odot\,yr^{-1}$ at $5 < z < 6$ has been estimated to be around $3 \times 10^{-6}\,Mpc^{-3}$ (ref. 24). Extrapolation then suggests that the SMG number density at $z \approx 7$ is approximately $1 \times 10^{-6}\,Mpc^{-3}$, which equates to roughly 1 SMG at $z \approx 7$ over the roughly 400-arcmin$^2$ area from which GS-9209 and one other $z > 4$ quiescent galaxy were selected[12]. This broadly consistent number density suggests that it is entirely plausible that GS-9209 went through a SMG phase at $z \approx 7$, shortly before quenching.

In Fig. 3b, we show the positions of the massive, high-redshift galaxy candidates recently reported by Labbe et al.[6] in the first imaging release from the JWST Cosmic Evolution Early Release Science (CEERS) Survey. The positions of these galaxies are broadly consistent with the SFH of GS-9209 at $z \approx 7$–8. It should however be noted that, as previously discussed, GS-9209 was selected as one of only two robustly identified $z > 4$ massive quiescent galaxies in an area roughly 10 times the size of the initial CEERS Survey imaging area[12]. It therefore seems unlikely that

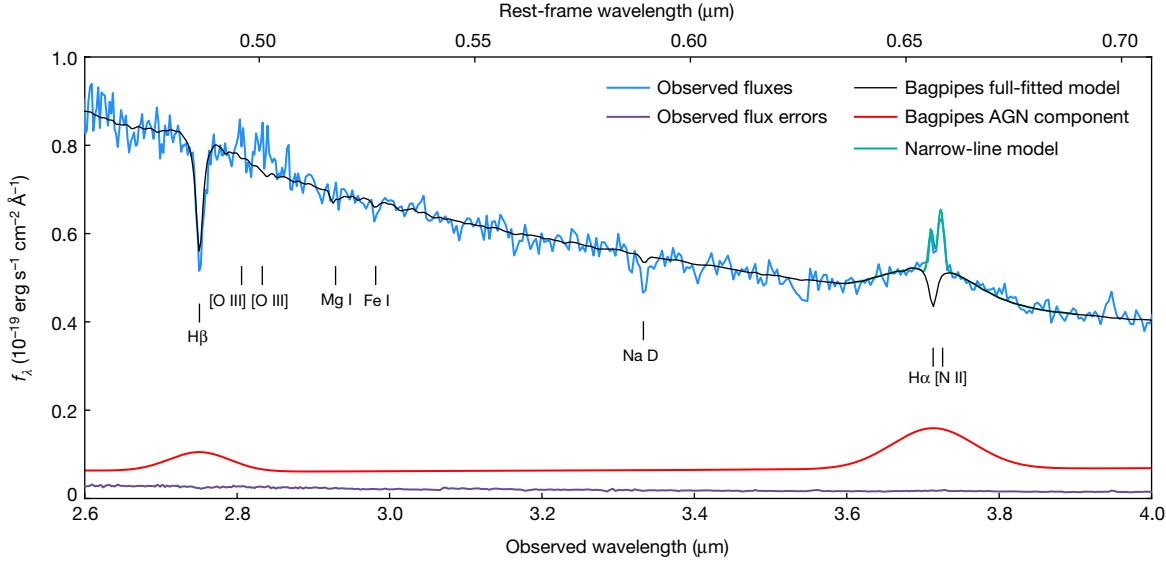

**Fig. 2 | JWST NIRSpec observations of GS-9209 with a zoom in on H$\beta$ and H$\alpha$.** Data are shown in blue, with their associated (1$\sigma$) uncertainties visible at the bottom in purple. The full Bagpipes fitted model is shown in black, with the AGN component shown in red. The narrow H$\alpha$ and [N II] lines were masked during the Bagpipes fitting process, and subsequently fitted with Gaussian functions, shown in green. Key emission and absorption features are also marked.

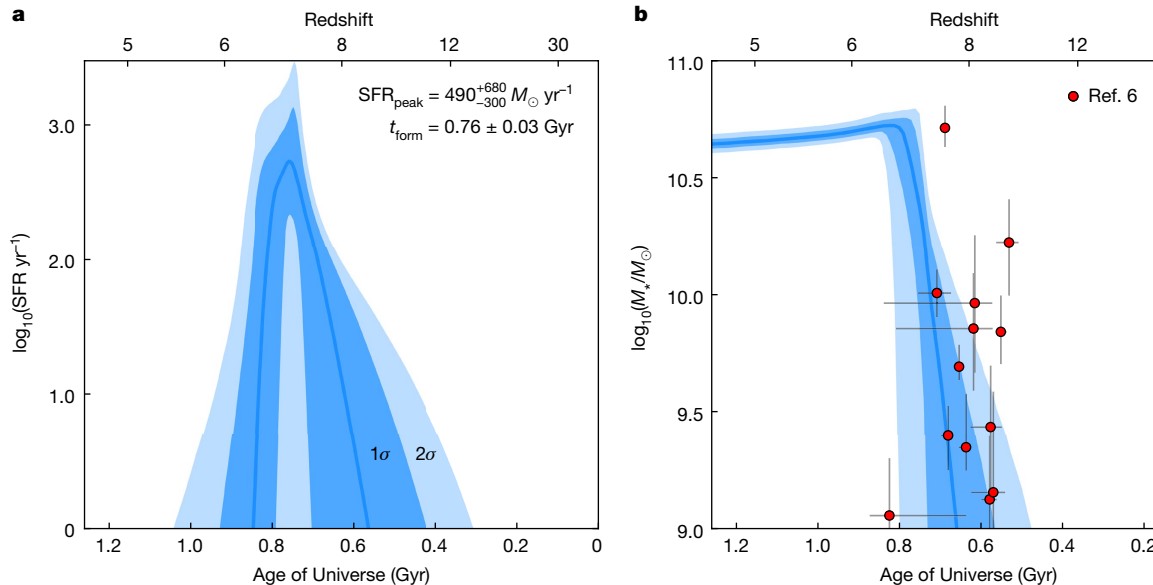

**Fig. 3 | The SFR and stellar mass of GS-9209 as a function of time. a**, The SFR as a function of time (the SFH). **b**, The stellar mass as a function of time. The blue lines show the posterior medians, with the darker and lighter shaded regions showing the 1$\sigma$ and 2$\sigma$ confidence intervals, respectively. We find a formation redshift, $z_{form} = 6.9 \pm 0.2$ and a quenching redshift, $z_{quench} = 6.5^{+0.2}_{-0.5}$.

The sample of massive $z \approx 8$ galaxy candidates from the JWST CEERS Survey reported by Labbe et al.[6] is also shown in **b**, demonstrating that these candidates are plausible progenitors for GS-9209. The uncertainties shown on the red points are 1$\sigma$ standard deviation values.

a large fraction of the candidates reported by Labbe et al.[6] will evolve in a way similar to that of GS-9209 over the redshift interval $z \approx 5$–8.

From our Bagpipes full spectral fit, we measured an observed broad H$\alpha$ flux of $f_{H\alpha,broad} = 1.26 \pm 0.08 \times 10^{-17}$ = erg s$^{-1}$ cm$^{-2}$ and full width at half maximum (FWHM) of 10,300 $\pm$ 700 km s$^{-1}$ in the rest frame. This linewidth, while very broad, is consistent with rest-frame ultraviolet broad linewidths measured for some $z \approx 6$ quasars[25,26].

As visualized in Fig. 2, we fitted Gaussian components to the narrow H$\alpha$ and [N II] lines following subtraction of our best-fitting Bagpipes model (Methods). We obtained a H$\alpha$ narrow-line flux of $1.58 \pm 0.10 \times 10^{-18}$ erg s$^{-1}$ cm$^{-2}$ and a [N II] flux of $1.56 \pm 0.10 \times 10^{-18}$ erg s$^{-1}$ cm$^{-2}$, giving a line ratio of $\log_{10}([N II]/H\alpha) = -0.01 \pm 0.04$. This line ratio is substantially higher than would be expected as a result of ongoing star formation, and is consistent with excitation due to an AGN or shocks resulting from galactic outflows[27]. Such outflows are commonly observed in post-starburst galaxies at $z \gtrsim 1$ (ref. 28). We discuss what can be learned about the SFR of GS-9209 from the observed H$\alpha$ flux in Methods.

We estimated the black-hole mass for GS-9209, $M_\bullet$, from our combined H$\alpha$ flux and broad-line width, using the relation presented in equation 6 of Greene and Ho[29], obtaining $\log_{10}(M_\bullet/M_\odot) = 8.7 \pm 0.1$. From our Bagpipes full spectral fit, we inferred a stellar velocity dispersion of $\sigma = 247 \pm 16$ km s$^{-1}$ for GS-9209, after correcting for the

intrinsic dispersion of our template set and instrumental dispersion. Given this measurement, the relationship between velocity dispersion and black-hole mass presented by Kormendy and Ho[30] predicts $\log_{10}(M_\bullet/M_\odot) = 8.9 \pm 0.1$.

Given the broad agreement between these estimators, it seems reasonable to conclude that GS-9209 contains a supermassive black hole with a mass of approximately half a billion to a billion solar masses. It is interesting to note that this is $\simeq 4$–5 times the black hole mass that would be expected given the stellar mass of the galaxy, assuming this is equivalent to the bulge mass. This is consistent with the observed increase in the average black-hole to bulge mass ratio for massive galaxies from $0 < z < 2$ (ref. 31). The large amount of historical AGN accretion implied by this substantial black-hole mass suggests that AGN feedback may have been responsible for quenching this galaxy[32].

GS-9209 is an extremely compact source, which is only marginally resolved in the highest-resolution available imaging data. We measured the size of GS-9209 using newly available JWST Near-Infrared Camera (NIRCam) F210M-band imaging, which has a FWHM of around 0.07″ (Methods). Accounting for the AGN point-source contribution, we measured an effective radius, $r_e = 0.033 \pm 0.003$″ for the stellar component of GS-9209, along with a Sérsic index, $n = 2.3 \pm 0.3$. At $z = 4.658$, this corresponds to $r_e = 215 \pm 20$ parsecs. This is consistent with previous

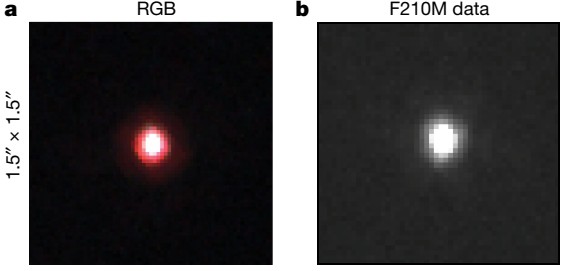

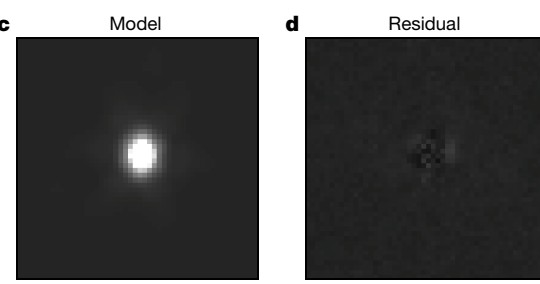

**Fig. 4 | JWST NIRCam imaging of GS-9209.** Each cut-out image shows an area of 1.5″ × 1.5″. **a**, RGB image, constructed with F430M as red, F210M as green and F182M as blue. **b**,**c**, The F210M image (**b**), with our posterior median PetroFit model shown in **c**. **d**, The residuals between model and data, on the same colour scale as **b** and **c**.

measurements by the Cosmic Assembly Near-infrared Deep Extragalactic Legacy Survey (CANDELS)/3D-HST team[33], and is about 0.7 dex below the mean relationship between $r_e$ and stellar mass for quiescent galaxies at $z \approx 1$ (refs. 33,34). This is interesting given that post-starburst galaxies at $z \approx 1$ are known to be more compact than is typical for the wider quiescent population[35]. We calculate a stellar-mass surface density within $r_e$ of $\log_{10}(\Sigma_{eff}/M_\odot \, \mathrm{kpc}^{-2}) = 11.15 \pm 0.08$, consistent with the densest stellar systems in the Universe[36]. We show the F210M data for GS-9209, along with our posterior median model, in Fig. 4.

We estimated the dynamical mass using our size and velocity dispersion measurements[28], obtaining a value of $\log_{10}(M_{dyn}/M_\odot) = 10.3 \pm 0.1$. This is about 0.3-dex lower than the stellar mass we measure. As GS-9209 is only marginally resolved, even in JWST imaging data, and owing to the presence of the AGN component, it is plausible that our measured $r_e$ may be subject to systematic uncertainties. Furthermore, because the pixel scale of NIRSpec is 0.1″, our velocity dispersion measurement may not accurately represent the central velocity dispersion, leading to an underestimated dynamical mass. It should also be noted that the stellar mass we measure is strongly dependent on our assumed IMF. A final, intriguing possibility would be a high level of rotational support in GS-9209, as has been observed for quiescent galaxies at $2 < z < 3$ (ref. 37). Unfortunately, the extremely compact nature of the source makes any attempt at resolved studies extremely challenging, even with the JWST NIRSpec integral field unit. Resolved kinematics for this galaxy would be a clear use case for the High Angular Resolution Monolithic Optical and Near-infrared Integral field spectrograph (HARMONI) planned for the Extremely Large Telescope (ELT).

GS-9209 demonstrates unambiguously that massive galaxy formation was already well underway within the first billion years of cosmic history and that the earliest onset of galaxy quenching was no later than around 800 Myr after the Big Bang. On the basis of the properties we measured, GS-9209 seems likely to be associated with the most extreme galaxy populations now known at $z > 5$, such as the highest-redshift SMGs and quasars[26,38,39]. GS-9209 and similar objects[8] are also likely progenitors for the dense, ancient cores of the most massive galaxies in the local Universe.

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

## Methods

### Spectroscopic data and reduction

The spectroscopic data shown in Fig. 1 were taken on 16 November 2022. The target was acquired directly by means of Wide Aperture Target Acquisition (WATA), meaning the object is extremely well centred. Spectroscopic observations were taken through the NIRSpec fixed slit (S200A1), which has a width of 0.2″. Data were taken using the G235M and G395M gratings, providing an average spectral resolution of $R = 1,000$. With each grating, a total of five integrations were taken at different dither positions along the slit. The read-out pattern used was NRSIRS2, with 30 and 20 groups per integration for the two gratings respectively, providing total integration times of 3 hours and 2 hours, respectively.

We reduced our NIRSpec data with the JWST Science Calibration Pipeline v.1.8.4, using v.1017 of the JWST calibration reference data. To improve the spectrophotometric calibration of our data, we also reduced observations of the A-type standard star 2MASS J18083474+6927286 (ref. 43), taken as part of JWST commissioning programme 1128 (principal investigator: Lützgendorf)[44] using the same instrument modes. We compared the resulting stellar spectrum against a spectral model for this star from the CALSPEC library[45] to construct a calibration function, which we then applied to our observations of GS-9209. The resulting spectrophotometry is well matched with the available near-infrared photometric data, and the calibration polynomial we fitted along with our Bagpipes model results only in further calibration changes at roughly the 10% level. We also find that the uncertainties output by the pipeline are only moderately underestimated, with the error bar expansion term in our Bagpipes model resulting in an increase of 50% to the pipeline-produced uncertainties, in agreement with other recent analyses (for example, https://github.com/spacetelescope/jwst/issues/7362).

### Photometric data reduction

Most of our photometric data were taken directly from the CANDELS GOODS South catalogue[46]. We supplemented these data with new JWST NIRCam photometric data taken as part of the Ultra Deep Field Medium-Band Survey[47] (Programme ID: 1963; PI: Williams). Data are available in the F182M, F210M, F430M, F460M and F480M bands. We reduced these data using the PRIMER Enhanced NIRCam Image-processing Library (PENCIL)[7], a custom version of the JWST Science Calibration Pipeline (v.1.8.0), and using v.1011 of the JWST calibration reference data. We measured photometric fluxes for GS-9209 in large, 1″-diameter apertures to ensure we measured the total flux in each band (the object is isolated, with no other sources within this radius; Fig. 4). We measured uncertainties as the standard deviation of flux values in the nearest 100 blank-sky apertures, masking out nearby objects[48].

### Bagpipes full spectral fitting

We fitted the available photometry in parallel with our new spectroscopic data using the Bagpipes code[18]. Our model has a total of 22 free parameters, describing the stellar, dust, nebular and AGN components of the spectrum. A full list of these parameters, along with their associated priors, is given in Extended Data Table 1. We fitted our model to the data using the MultiNest nested sampling algorithm[49–51]. The full Bagpipes fitted to our combined dataset, along with residuals, is shown in Extended Data Fig. 1. Posterior percentiles for our fit to the data are given in Extended Data Table 2.

We used the 2016 revised version of the BC03 (refs. 52,53) stellar population models, using the MILES stellar spectral library[54] and revised stellar evolutionary tracks[55,56]. We assumed a double power-law SFH model[18,19]. We allowed the logarithm of the stellar metallicity, $Z_*$, to vary freely from $\log_{10}(Z_*/Z_\odot) = -2.45$ to 0.55. These are the limits of the range spanned by the BC03 model grid relative to our adopted solar metallicity value ($Z_\odot = 0.0142$) (ref. 57).

We masked out the narrow emission lines in our spectrum during our Bagpipes fitting because of likely AGN contributions, whereas Bagpipes is capable of modelling emission lines only from star-forming regions. We did however still include a nebular model in our Bagpipes fit to allow for the possibility of nebular continuum emission from star-forming regions. We assumed a stellar birth cloud lifetime of 10 Myr, and varied the logarithm of the ionization parameter, $U$, from $\log_{10}(U) = -4$ to $-2$. We also allowed the logarithm of the gas-phase metallicity, $Z_g$, to vary freely from $\log_{10}(Z_g/Z_\odot) = -2.45$ to 0.55. Because our eventual fitted model includes only an extremely small amount of star formation in the past 10 Myr for GS-9209, this nebular component makes a negligible contribution to the fitted model spectrum.

We modelled attenuation of the above components by dust using the model of Noll et al.[58] and Salim et al.[59], which is parameterized as a power-law deviation from the Calzetti dust attenuation law[60], and also includes a Drude profile to model the 2,175-Å bump. We allowed the $V$-band attenuation, $A_V$ to vary from 0 to 4 magnitudes. We further assumed that attenuation is multiplied by an extra factor for all stars with ages below 10 Myr and resulting nebular emission. This factor is commonly assumed to be 2; however, we allowed it to vary from 1 to 5.

We allowed redshift to vary, using a narrow Gaussian prior with a mean of 4.66 and standard deviation of 0.01. We also convolved the spectral model with a Gaussian kernel in velocity space, to account for velocity dispersion in our target galaxy. The width of this kernel is allowed to vary with a logarithmic prior across a range of $50 – 500$ km s$^{-1}$. The resolution of our spectroscopic data is high enough that the total dispersion is dominated by stellar velocity dispersion within the target galaxy, which has a standard deviation of $\sigma \approx 250$ km s$^{-1}$, compared with the average instrumental dispersion of $\sigma \approx 128$ km s$^{-1}$.

Separately from the above components, we also included a model for AGN continuum, broad Hα and Hβ emission. Following Vanden Berk et al.[61], we modelled AGN continuum emission with a broken power law, with two spectral indices and a break at $\lambda_{rest} = 5,000$ Å in the rest frame. We varied the spectral index at $\lambda_{rest} < 5,000$ Å using a Gaussian prior with a mean value of $\alpha_\lambda = -1.5$ ($\alpha_v = -0.5$) and standard deviation of 0.5. We also varied the spectral index at $\lambda_{rest} > 5,000$ Å using a Gaussian prior with a mean value of $\alpha_\lambda = 0.5$ ($\alpha_v = -2.5$) and standard deviation of 0.5. We parameterized the normalization of the AGN continuum component using $f_{5100}$, the flux at rest-frame 5,100 Å, which we allowed to vary with a linear prior from 0 to $10^{-19}$ erg s$^{-1}$ cm$^{-2}$ Å$^{-1}$.

We modelled broad Hα with a Gaussian component, varying the normalization from 0 to $2.5 \times 10^{-17}$ erg s$^{-1}$ cm$^{-2}$ using a linear prior, and the velocity dispersion from 1,000 to 5,000 km s$^{-1}$ in the rest frame using a logarithmic prior. We also included a broad Hβ component in the model, which has the same parameters as the broad Hα line, but with normalization divided by the standard 2.86 ratio from Case B recombination theory. However, as shown in Fig. 2, this Hβ model peaks at around the noise level in our spectrum, and the line is therefore plausible in not being obviously detected in the observed spectrum.

We included intergalactic medium absorption using the model of Inoue et al.[62]. To allow for imperfect spectrophotometric calibration of our spectroscopic data, we also included a second-order Chebyshev polynomial[63–65], which the above components of our combined model were all divided by before being compared with our spectroscopic data. We finally fitted an extra white noise term, which multiplies the spectroscopic uncertainties from the JWST pipeline by a factor, $a$, which we vary with a logarithmic prior from 1 to 10.

### Investigation of alternative SFH models

The functional forms used to model galaxy SFHs are well known to substantially affect physical parameter inferences[19–21], with the degree of systematic uncertainty highly dependent on the physical parameter of interest, the type of data and the galaxy being studied. In this section, we test the dependence of our inferred formation and quenching times for GS-9209 on the SFH model used. We re-run our Bagpipes full

spectral fitting analysis, substituting the double power-law SFH model described above, first for the continuity non-parametric model[20], and second for a simple top-hat (constant) SFH model. For the continuity model, we use 8 time bins, with bin edges at 0, 10, 100, 200, 400, 600, 800, 1,000 and 1,260 Myr before observation. For the top-hat model, we vary the time at which star formation turned on with a uniform prior between the Big Bang and time of observation. We vary the time at which star formation then stopped with a uniform prior from the time at which star formation turned on to the time of observation.

The results of these alternative fitting runs are shown in Extended Data Fig. 2. This figure shows two alternative versions of Fig. 3, with the continuity non-parametric model results shown in panels a and b, and the top-hat model results shown in panels c and d. The SFH posteriors shown, while varying in their detailed shapes, are in good overall agreement with our original double power-law model. For the double power-law model, we recovered $t_{form} = 0.76 \pm 0.03$ Gyr and $t_{quench} = 0.83^{+0.08}_{-0.06}$ Gyr after the Big Bang. The values returned under the assumption of these other two models are consistent to within $1\sigma$. For the continuity non-parametric model, we recovered $t_{form} = 0.74^{+0.02}_{-0.03}$ Gyr and $t_{quench} = 0.86^{+0.19}_{-0.01}$ Gyr. For the top-hat model we recovered $t_{form} = 0.74 \pm 0.02$ Gyr and $t_{quench} = 0.91^{+0.04}_{-0.06}$ Gyr. Both of these models also produce stronger constraints on the peak historical SFR of GS-9209 at a lower level than the double power-law model, although still consistent within $1\sigma$. We conclude that our key results are not strongly dependent on our choice of SFH model.

### AGN contribution and fitting of narrow emission lines
From our Bagpipes full spectral fit, we recovered an observed AGN continuum flux at rest-frame wavelength $\lambda_{rest} = 5,100$ Å of $f_{5100} = 0.06 \pm 0.01 \times 10^{-19}$ erg s$^{-1}$ cm$^{-2}$ Å$^{-1}$. This is approximately 7.5% of the total observed flux from GS-9209 at $\lambda = 2.9$ μm. We measured a power-law index for the AGN continuum emission of $\alpha_\lambda = -0.5 \pm 0.3$ at $\lambda_{rest} < 5,000$ Å and $\alpha_\lambda = 0.4 \pm 0.3$ at $\lambda_{rest} > 5,000$ Å. The AGN contribution to the continuum flux from GS-9209 rises to around 10% at the blue end of our spectrum ($\lambda = 1.7$ μm), and around 20% at the red end ($\lambda = 5$ μm). Just above the Lyman break at $\lambda \approx 7,000$ Å, the AGN contribution is about 35% of the observed flux.

Given our measured $f_{H\alpha,broad}$, which is more direct than our AGN continuum measurement, the average relation for local AGN presented by Greene and Ho[29] predicts $f_{5100}$ to be roughly 0.2 dex brighter than we measure. However, given the intrinsic scatter of 0.2 dex that they report, our measured $f_{5100}$ is only $1\sigma$ below the mean relation. The extreme equivalent widths of the observed Balmer absorption features firmly disfavour stronger AGN continuum emission.

We fitted the narrow Hα and [N II] lines in our spectrum as follows. We first subtracted from our observed spectrum the posterior median Bagpipes model from our full spectral fitting. We then simultaneously fitted Gaussian components to both lines, assuming the same velocity width for both, which was allowed to vary. This process is visualized in Fig. 2. We also show the broad Hβ line in our AGN model, for which we assume the same width as broad Hα, as well as Case B recombination. It can be seen that the broad Hβ line peaks at around the noise level in our spectrum, and is hence too weak to be clearly observed in our data.

### The SFR of GS-9209
In this section, we discuss the available observational indicators for the SFR of GS-9209. The commonly applied sSFR threshold for defining quiescent galaxies is sSFR$_{threshold} = 0.2/t_H$, where $t_H$ is the age of the Universe[23]. For GS-9209 at $z = 4.658$ and $\log_{10}(M_*/M_\odot) \approx 10.6$, this corresponds to $\log_{10}(sSFR_{threshold}/yr^{-1}) \approx -9.8$, or SFR$_{threshold} \approx 6 \, M_\odot$ yr$^{-1}$.

In Santini et al.[66], the authors report that GS-9209 is undetected in the Atacama Large Millimeter/submillimeter Array (ALMA) band-6 data, with a flux of $-0.05 \pm 0.16$ mJy per beam, from which they derive a $1\sigma$ upper limit on SFR of $41 \, M_\odot$ yr$^{-1}$. They also perform a stacking experiment, with stacked ALMA band-6 data for a sample of 20 objects

selected as $3 < z < 5$ quiescent galaxies (including GS-9209) still yielding no detection, implying that the average SFR for this sample is well below the individual-object detection limit. The extremely blue spectral shape of this object in the rest-frame red-optical to near-infrared (observed frame 2–8 μm; Extended Data Fig. 1) is also consistent with no substantial obscured star-forming or AGN component. Deeper ALMA data for this object would be of value for setting a more stringent direct upper bound on obscured star formation.

As discussed in the main text, the high [N II]/Hα ratio in our observed spectrum strongly suggests that this line emission is powered by AGN activity or shocks. However, if we assume that all the narrow Hα emission is driven by continuing star formation, neglecting dust attenuation, we obtain SFR = $1.9 \pm 0.1 \, M_\odot$ yr$^{-1}$ (ref. 67), corresponding to $\log_{10}(sSFR/yr^{-1}) = -10.3 \pm 0.1$. Measurements of the average dust attenuation on Hα emission, $A_{H\alpha}$, are not yet available at $z \approx 5$; however, from $0 < z < 2$, stellar mass is found to be the most important factor in predicting the level of dust attenuation[68,69], with little evolution observed across this redshift interval. At $z \approx 2.3$, the average $A_{H\alpha}$ for galaxies with $\log_{10}(M_*/M_\odot) \approx 10.6$ is 1.25 magnitudes[69], which would suggest that the SFR of GS-9209 is roughly $6 \, M_\odot$ yr$^{-1}$. However, given the multiple lines of evidence we uncovered for a substantial non-stellar component to the Hα line, combined with the fact that the extremely low stellar continuum $A_V$ implies that the gas-phase attenuation is also low[70], it is probable that the sSFR of GS-9209 is considerably lower than the threshold normally applied for selecting quiescent galaxies.

### Size measurement from F210M-band imaging
The CANDELS/3D-HST team[33] measured an effective radius of $r_e = 0.029 \pm 0.002''$ for GS-9209 in the HST F125W filter by means of Sérsic fitting, along with a Sérsic index, $n = 6.0 \pm 0.8$. At $z = 4.658$, this corresponds to $r_e = 189 \pm 13$ parsecs. We revised this measurement using the newly acquired JWST NIRCam F210M imaging data discussed above. We modelled the light distribution of GS-9209 using PetroFit[71]. We fitted these PetroFit models to our data using the MultiNest nested sampling algorithm[49–51]. We used F210M in preference to the F182M band due to the smaller AGN contribution in F210M and the fact that it sits above the Balmer break, thereby being more representative of the stellar mass present rather than any continuing star formation.

As our spectroscopic data contain strong evidence for an AGN, we fitted both Sérsic and delta-function components simultaneously, convolved by an empirically estimated point spread function (PSF), derived by stacking bright stars. In preliminary fitting, we found that the relative fluxes of these two components are entirely degenerate with the Sérsic parameters. We therefore predicted the AGN contribution to the flux in this band on the basis of our full spectral fitting result, obtaining a value of 8% ± 1%. We then imposed this as a Gaussian prior on the relative contributions from the Sérsic and delta-function components. The 11 free parameters of our model are the overall flux normalization, which we fitted with a logarithmic prior; the effective radius, $r_e$; Sérsic index, $n$; ellipticity and position angle of the Sérsic component; the $x$ and $y$ centroids of both components; the position angle of the PSF; and the fraction of light in the delta-function component, which we fitted with a Gaussian prior with a mean of 8% and standard deviation of 1%, on the basis of our full spectral fitting result.

Deeper imaging data in the F200W or F277W bands (for example, from the JWST Advanced Deep Extragalactic Survey) will provide a useful check on our size measurement for GS-9209, particularly given the lower AGN fraction in the F277W band.

### Data availability
The datasets analysed during the current study are available from the Mikulski Archive for Space Telescopes (MAST) repository at https://mast.stsci.edu. The spectrum for GS-9209 was observed under JWST Programme ID 2285 (principal investigator: Carnall). This programme

has a 12-month proprietary period, and data will automatically become publicly available through MAST on 16 November 2023. Reduced data products are available from the corresponding author upon request.

## Code availability

The Bagpipes code is publicly available at https://github.com/ACCarnall/bagpipes. The PetroFit code is publicly available at https://github.com/PetroFit/petrofit. The JWST data reduction pipeline is publicly available at https://github.com/spacetelescope/jwst.

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

**Acknowledgements** We would like to thank J. Aird for helpful discussions. A.C.C. thanks the Leverhulme Trust for their support through a Leverhulme Early Career Fellowship. R.J.M., J.S.D., D.J.M., V.W., R.B., C.T.D. and M.L.H. acknowledge the support of the Science and Technology Facilities Council. F.C. acknowledges support from a UK Research and Innovation Frontier Research Guarantee Grant (grant reference EP/X021025/1). A.C. acknowledges support from the grant PRIN MIUR 2017 - 20173ML3WW_001.

**Author contributions** A.C.C. led the preparation of the observing proposal, reduction and analysis of the data, and preparation of the manuscript. R.J.M., J.S.D., V.W., F.C. and A.C. provided advice and assistance with data reduction, analysis and interpretation, and consulting on the preparation of the observing proposal. D.J.M., D.M., R.B. and C.T.D. reduced the JWST imaging data and prepared the empirical PSF. D.J.M., M.L.H. and S.M.J. assisted with measurement of the size and morphology of GS-9209. S.W. assisted with selection of GS-9209 from the CANDELS catalogues before the observing proposal was submitted. All authors assisted with preparation of the final published manuscript.

**Competing interests** The authors declare no competing interests.

**Additional information**
**Correspondence and requests for materials** should be addressed to Adam C. Carnall.

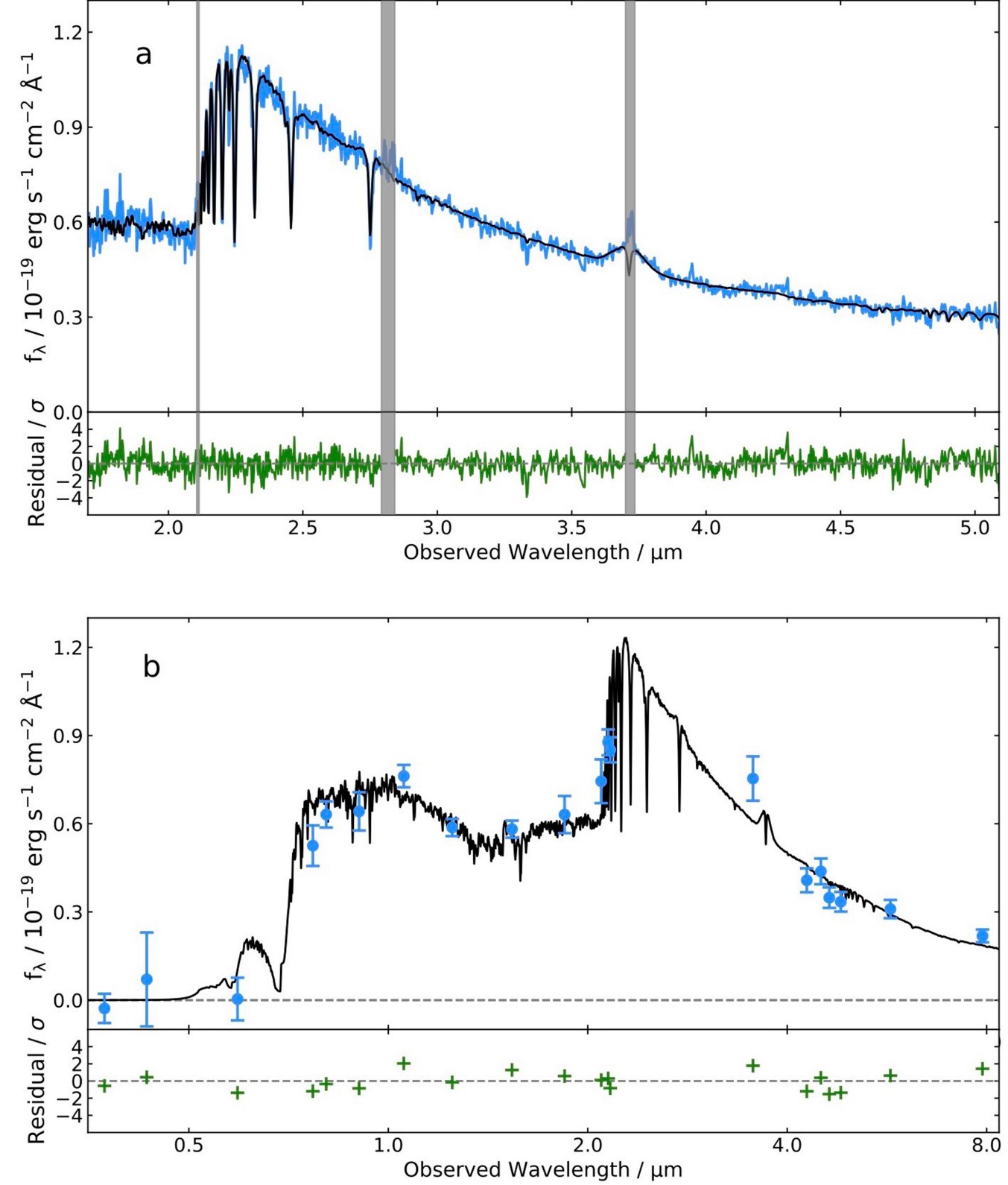

**Extended Data Fig. 1 | Model fit to our full combined dataset.** Panel a shows our spectroscopic dataset in blue, with the full Bagpipes fitted model shown in black, and residuals between model and data shown below in green. Regions of our spectroscopic dataset masked during our Bagpipes fitting are shaded gray. Panel b shows our photometric data, again in blue, with the full model again shown in black, and residuals below marked with green crosses.

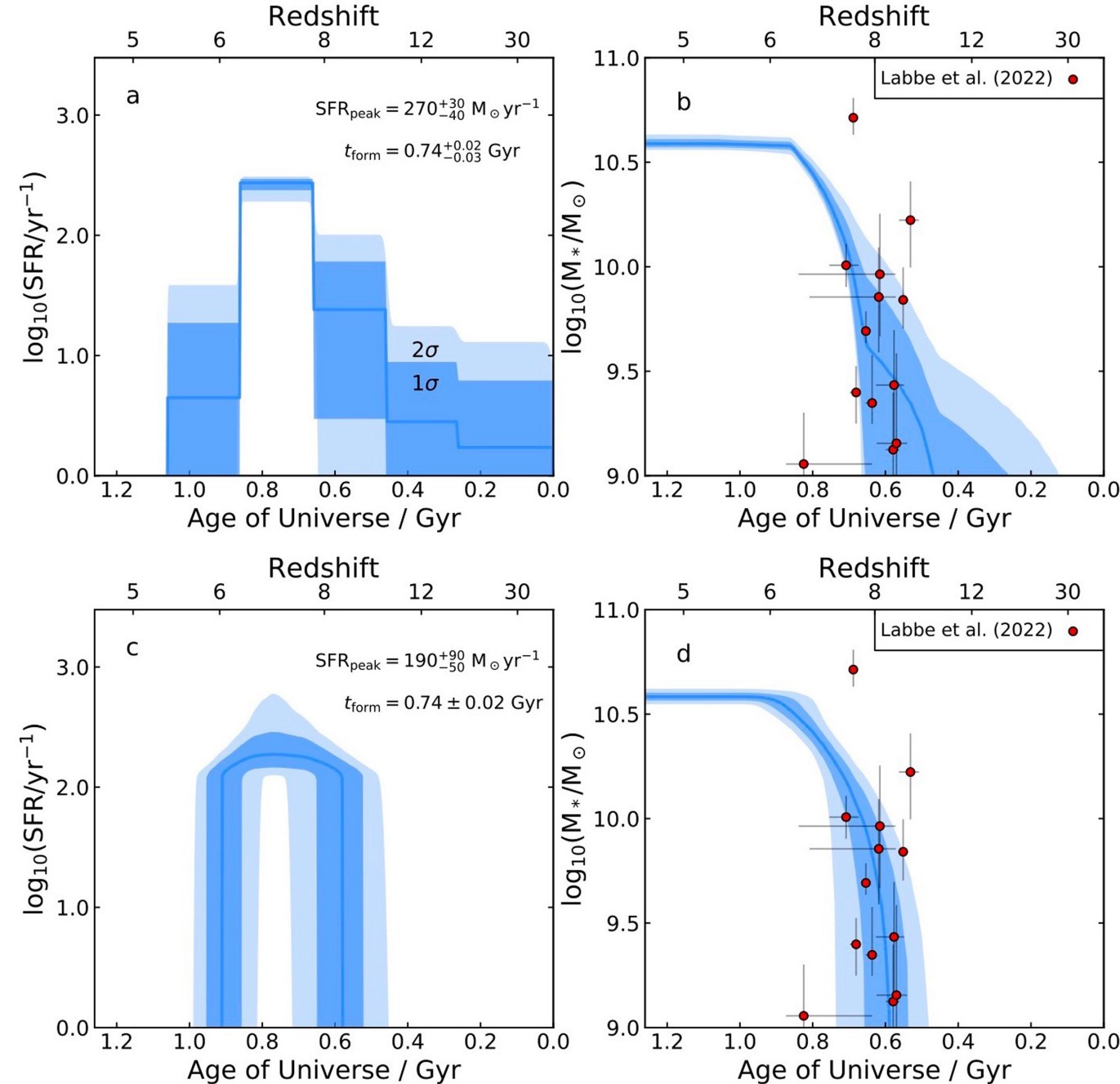

**Extended Data Fig. 2 | The star-formation rate and stellar mass of GS-9209 as a function of time under the assumption of different models.** Panels a and b show the star-formation rate and stellar mass respectively as a function of time under the assumption of the continuity non-parametric SFH model[20] (as an alternative to the double-power-law model used in the main analysis and shown in Fig. 3). Panels c and d again show the star-formation rate and stellar mass respectively as a function of time, this time assuming a top-hat SFH model. Consistent formation times are recovered using all three models.

**Extended Data Table 1 | The 22 free parameters of the Bagpipes model we fit to our spectroscopic and photometric data, along with their associated prior distributions**

| Component | Parameter | Symbol / Unit | Range | Prior | Hyper-parameters | |
|---|---|---|---|---|---|---|
| General | Redshift | $z$ | (4.6, 4.7) | Gaussian | $\mu = 4.66$ | $\sigma = 0.01$ |
| | Stellar velocity dispersion | $\sigma$ / km s$^{-1}$ | (50, 500) | Logarithmic | | |
| SFH | Total stellar mass formed | $M_*$ / M$_\odot$ | (1, $10^{13}$) | Logarithmic | | |
| | Stellar metallicity | $Z_*$ / Z$_\odot$ | (0.00355, 3.55) | Logarithmic | | |
| | Double-power-law falling slope | $\alpha$ | (0.01, 1000) | Logarithmic | | |
| | Double-power-law rising slope | $\beta$ | (0.01, 1000) | Logarithmic | | |
| | Double-power-law turnover time | $\tau$ / Gyr | (0.1, $t_{\rm obs}$) | Uniform | | |
| Dust | $V-$band attenuation | $A_V$ / mag | (0, 4) | Uniform | | |
| | Deviation from Calzetti slope | $\delta$ | (−0.3, 0.3) | Gaussian | $\mu = 0$ | $\sigma = 0.1$ |
| | Strength of 2175Å bump | $B$ | (0, 5) | Uniform | | |
| | Attenuation ratio for birth clouds | $\epsilon$ | (1, 5) | Uniform | | |
| AGN | Power law slope ($\lambda < 5000$ Å) | $\alpha_{\lambda < 5000\text{Å}}$ | (−2, 2) | Gaussian | $\mu = -1.5$ | $\sigma = 0.5$ |
| | Power law slope ($\lambda > 5000$ Å) | $\alpha_{\lambda > 5000\text{Å}}$ | (−2, 2) | Gaussian | $\mu = 0.5$ | $\sigma = 0.5$ |
| | H$\alpha$ broad-line flux | $f_{\text{H}\alpha,\,\text{broad}}$ / erg s$^{-1}$ cm$^{-2}$ | (0, $2.5 \times 10^{-17}$) | Uniform | | |
| | H$\alpha$ broad-line velocity dispersion | $\sigma_{\text{H}\alpha,\,\text{broad}}$ / km s$^{-1}$ | (1000, 5000) | Logarithmic | | |
| | Continuum flux at $\lambda = 5100$ Å | $f_{5100}$ / erg s$^{-1}$ cm$^{-2}$ Å$^{-1}$ | (0, $10^{-19}$) | Uniform | | |
| Nebular | Ionization parameter | $U$ | ($10^{-4}$, $10^{-2}$) | Logarithmic | | |
| | Gas-phase metallicity | $Z_g$ / Z$_\odot$ | (0.00355, 3.55) | Logarithmic | | |
| Calibration | Zero order | $P_0$ | (0.75, 1.25) | Gaussian | $\mu = 1$ | $\sigma = 0.1$ |
| | First order | $P_1$ | (−0.25, 0.25) | Gaussian | $\mu = 0$ | $\sigma = 0.1$ |
| | Second order | $P_2$ | (−0.25, 0.25) | Gaussian | $\mu = 0$ | $\sigma = 0.1$ |
| Noise | White noise scaling | $a$ | (0.1, 10) | logarithmic | | |

The upper limit on $\tau$, $t_{\rm obs}$, is the age of the Universe as a function of redshift. Logarithmic priors are all applied in base ten. For parameters with Gaussian priors, the mean is $\mu$ and the standard deviation is $\sigma$.

**Extended Data Table 2 | Posterior percentiles for the 22 free parameters of the Bagpipes model we fit to our spectroscopic and photometric data**

| Parameter | 16$^{\text{th}}$ Percentile | 50$^{\text{th}}$ percentile | 84$^{\text{th}}$ Percentile |
|---|---|---|---|
| $z$ | 4.6580 | 4.6582 | 4.6584 |
| $\sigma$ / km s$^{-1}$ | 256 | 269 | 283 |
| $\log_{10}(M_* / M_\odot)$ | 10.81 | 10.83 | 10.85 |
| $Z_* / Z_\odot$ | 0.09 | 0.11 | 0.12 |
| $\alpha$ | 42 | 155 | 511 |
| $\beta$ | 10 | 22 | 113 |
| $\tau$ / Gyr | 0.71 | 0.75 | 0.79 |
| $A_V$ / mag | 0.00 | 0.02 | 0.04 |
| $\delta$ | $-0.05$ | 0.00 | 0.05 |
| $B$ | 0.81 | 2.47 | 4.15 |
| $\epsilon$ | 1.78 | 2.98 | 4.22 |
| $\alpha_{\lambda<5000\text{Å}}$ | $-0.79$ | $-0.52$ | $-0.27$ |
| $\alpha_{\lambda>5000\text{Å}}$ | 0.14 | 0.40 | 0.72 |
| $f_{\text{H}\alpha,\text{broad}}$ / erg s$^{-1}$ cm$^{-2}$ | $2.8 \times 10^{-20}$ | $3.4 \times 10^{-20}$ | $3.9 \times 10^{-20}$ |
| $\sigma_{\text{H}\alpha,\text{broad}}$ / km s$^{-1}$ | 4100 | 4400 | 4700 |
| $f_{5100}$ / erg s$^{-1}$ cm$^{-2}$ Å$^{-1}$ | $1.18 \times 10^{-17}$ | $1.26 \times 10^{-17}$ | $1.35 \times 10^{-17}$ |
| $\log_{10}(U)$ | -3.65 | -3.00 | -2.37 |
| $Z_g / Z_\odot$ | 0.02 | 0.17 | 1.54 |
| $P_0$ | 1.13 | 1.15 | 1.18 |
| $P_1$ | 0.09 | 0.1 | 0.12 |
| $P_2$ | $-0.08$ | $-0.07$ | $-0.06$ |
| $a$ | 1.49 | 1.51 | 1.53 |

Full definitions of these parameters, along with their associated prior distributions, are given in Extended Data Table 1.