## [Peer Review File · Nature]

Manuscript Title: A massive quiescent galaxy at redshift 4.658

Reviewer Comments & Author Rebuttals

Reviewer Reports on the Initial Version:

Referee #1 (Remarks to the Author):

This is a well-written paper on the analysis of a high quality spectrum with JWST of the earliest galaxy with an evolved stellar population. While it is difficult to learn much definitively from a single object, and similar investigations have been published with similar outcomes (e.g., Glazebrook+17, Forrest+20), the galaxy presented here is the most distant yet, and the spectrum is of extraordinary quality. As such its findings represent a leap beyond earlier work, are potentially significant, and in my opinion deserving of consideration by Nature.

While the analysis is quite thorough and I support the overall findings, I have concerns about parts of the analysis that I would like to see addressed, specifically about the star formation history and residual star formation, AGN contribution, and in particular about the inferences on metallicity, which seems to me the weakest part. I also think more information about the methods are needed, as some aspects of the calibration, including absolute flux calibration and resolution, are still in their early days for JWST and may impact the results.

1) Star formation histories. These are sensitive to the parameterization of the SFH, as I know the authors are well aware of. It raises the question whether the peaked formation time and other aspects of the formation history are driven by the choice of SFH model. At least some different parameterizations should be explored: for example a top-hat with cut-off, or perhaps a "non-parametric" model, which are of course still parametric, but more flexible.

This would certainly imply less precise constraints on the formation and quenching timescale, and possibly even stellar mass. All of these quantities are reported at percent level accuracy, which are difficult to believe. Likely, the full uncertainty is not captured by sampling the posterior, but hidden in the variety of assumptions adopted in the modeling.

Related, I was wondering to what level obscured star formation can be ruled out, presumably from the ALMA limits. The limit on SFR from H-alpha does not take into account dust. It is argued from the stellar A_V that this should be small, but the star forming component could be obscured differently than the stellar continuum.

What would H-alpha corrected for nebular extinction using the (limit on the) Balmer decrement be? The depth of H-beta may be consistent with a large Balmer decrement. I presume this was not explored because the emission lines and continuum were not jointly fit. Some attempt could be made. The authors can still argue effectively that the high NII/Halpha is better explained by AGN activity or shocks, but it is important to give a full accounting of the possibility of any obscured on-going star formation.

2) Fe/H. This part of the analysis seems curiously weak compared to the rest of the analysis. Without any clear (absorption line) tracer of metallicity the entire analysis depends on constraints from the continuum shape over a rather small wavelength range. This seems tenuous and the implications for metallicity evolution seem not well supported. At the same time, a somewhat higher metallicity would weaken the (already limited) evidence for alpha-enhancement.

Could this be degenerate with many other factors that subtly affect the continuum slope? From absolute flux calibration of the instrument, and polynomial correction, which is not discussed quantitatively in the paper; to the contribution of the AGN continuum, which is subdominant but not negligible; and lastly, perhaps most importantly, the choice of stellar population itself.

If the authors really wish to make a strong case, it may be necessary to explore stellar population models beyond Bagpipes / BC03. Otherwise I think this aspect should be toned down, without further affecting the impact of the paper.

3) AGN. I am wondering how well the continuum power law slopes are constrained. I am presuming these parameters were jointly fit, together with the rest of the stellar population model. It struck me that the priors were fairly narrow and that the solutions fell close to the peak of the priors, which suggests to me that it may not be so well constrained by the observations. Are the contributions robust against using a more permissible, e.g., a wider gaussian or even uniform prior? I suspect the continuum contribution could be higher and the shape different. The current reasoning based on the depth of H-beta absorption seems qualitative. Perhaps it does not affect the inferred SFHs much, but it would be good to know. It may lead to more realistic uncertainties.

I am a bit in the dark, to be honest, on how evidence for a modestly over-massive black hole amounts to strong implications that AGN feedback is responsible for quenching the galaxy. I do not understand the argument that this implies causality.

4) Methods: More information should be provided on data reduction, calibration, and some aspects of the modeling. These are still early days of JWST and calibration is preliminary. And while the spectra are of impressive quality, it could well be that systematic errors are dominating the accuracy of the analysis. I suspect that the currently reported uncertainties are optimistic.

Some more information should be provided on the details of the mode, centering of the object in the slit (presumably very good), more detail on the absolute flux calibration of the instrument and proposed uncertainty. Comparison to e.g. NIRCcam photometry, although this has its limitations too. A polynomial correction was applied. How big was it. Does this correction affect the measurements significantly? In addition, the pipeline is formally calibrated to a uniformly exposed slitlet, whereas this source is effectively a point source. The resolution of the NIRSpec g235m and g295m is not $R=1000$, but varies by a factor of almost two across the wavelength range. Some of this may work in the authors advantage, in the sense that the resolution is high enough that the results do not depend on detailed treatment, but it would be good to at least discuss. How big was the noise correction?

Smaller comments:

1) The comparison with the sample reported by Labbe+23 is intriguing, albeit speculative until that sample is spectroscopically confirmed. Are these masses all reported on the same IMF?

2) It would be nice to include measurements on other important lines, specifically [OIII]4960,5008, as this line seems clearly detected, and may provide more clues about the ionization mechanism.

3) I am not sure if the IR limits give strong constraints on the amount of obscured AGN activity. I suspect the observational limits are fairly weak, given the high redshift and high expected dust temperature.

4) I presume the difference between the measured sersic index from F125W and this work at F210M is related to the fact that the AGN may contribute more at shorter wavelengths, and that accounting for any contribution in a point source would tend to lead to lower sersic n ? How does not including (or e.g. a 20% contribution) impact the fit?

I am assuming that accounting for spectral resolution does not play a significant role in the error budget? (see notes on methods). The low dynamical mass is somewhat surprising, although in line with work on dynamical masses of quiescent galaxies at $z=2-4$. If these galaxies end up in the cores (e.g. van Dokkum+17), the expected IMF might be bottom heavy and the discrepancy would become worse compared to the Kroupa IMF assumed here. But I agree that for objects this compact, resolved (lensed) studies are needed until ELTs come online to address this issue.

5) A nice overview of the parameters and assumptions is provided. Perhaps a similar overview could be provided for the best fit parameters.

6) Some display of the fit residuals would also be welcome.

Referee #2 (Remarks to the Author):

I have read the manuscript "A Massive Quiescent Galaxy at redshift 4.658" by Carnall et al. These authors report new JWST observations of the galaxy GS-9209, which reveal a number of remarkable features. The paper is well-written and the results are both novel and important for our understanding of galaxy formation in the first few hundred Myr of the Universe. I have only one major comment and several minor suggestions which I ask the authors to address.

Major:

1. I do not believe the authors estimate of the alpha-enhancement of GS-9209. The authors correctly note that both NaD and CaK are very likely affected by absorption by cold gas within the galaxy (whether in the ISM or in an outflow). The fact that NaD absorption is so much stronger than the models argues that NaD is compromised by intervening cold gas, and there is every reason to

assume that CaK is similarly affected. It is therefore simply not possible to use those features to estimate $[a/Fe]$. I therefore strongly urge the authors to remove discussion of $[a/Fe]$ from the paper.

Minor:

1. The authors use the word "extreme" and similar emphatic words too liberally. I suggest a close edit of the paper to tone down some of the strong wording - the extraordinary nature of this object is manifestly obvious!
2. Do the derived stellar population properties change if both NaD and CaK are masked in the BagPipes fitting? I'd like to know if the SFH and mass change by more than 1 sigma if these regions are excluded from the fit.
3. Many quantities reported in this paper have error bars that seem unrealistically small (for example the stellar mass with uncertainty of 0.02 dex and BH mass with uncertainty 0.1 dex). I realize the quoted errors are only statistically, and the authors surely appreciate that systematic uncertainties are likely much larger. At minimum, I would like the authors to add a bit of text here and there emphasizing that the uncertainties are stat-only and systematic uncertainties are likely much larger.
4. I would like to see the fit to the photometric SED in the Methods section.
5. As the authors are likely aware, the empirical MILES library contains very few hot metal-poor stars so one should be careful about using the spectral shape of A stars to estimate metallicity from models based on MILES stars. If the authors have access to theoretical spectral libraries, I would encourage them to try fitting with those libraries to see if a similarly low metallicity is recovered when fitting GS-9209. [This comment is somewhere in between "major" and "minor"....]

- Charlie Conroy

Author Rebuttals to Initial Comments:

The authors thank the referees and editor for their consideration, and for their recommendations to improve the paper. In response, we have made extensive revisions to the style of the paper, following the editor's guidance, to meet with journal style. The three key changes that have been made in response to the comments from the scientific referees are a.) the removal of the discussion of stellar metallicity, b.) the inclusion of tests demonstrating the effects of different star-formation history model choices on our results, and c.) the broadening of the priors on the AGN continuum slopes, which leads to very minor changes to our derived parameter values. We provide a full response to the referee and editorial comments below. Changes to the paper have been highlighted in red in the revised draft.

>>> Neither referee thinks that the metallicity arguments are much value, and both comment that the uncertainties are unrealistically small.

The authors agree with the referees that this is the weakest section of the analysis we present. Given this, and the request from the editor to shorten the paper, we have removed the discussion of stellar metallicity.

>>> Editorially, the paper is in rather poor shape. It is far too long, and not remotely close to style. I wish that you had contacted me in advance of submission so we could have sorted out those issues ahead of time. In addition, the paper is full of blatant advertising, which must be entirely removed. By advertising I mean statements like "new redshift record", "new deep continuum", etc. The abstract as it stands should be removed, and below I will offer guidance on crafting one that is in Nature's style.

The paper has been edited for Nature style, as described in more detail in our responses to the below points. We hope the editor finds that the new draft meets their requirements, however we would be happy to further iterate on this if more adjustments are needed.

>>> Please note that the presentation of a Nature paper has undergone significant changes in recent years to improve its readability and navigability online. Specifically, in most instances any supplementary text and data figures/tables can now be integrated into the main paper rather than presented in a separate Supplementary Information file. An overview of the key features and differences may be found in the Composition of a Nature Paper (http://s3-service-broker-live-19ea8b98-4d41-4cb4-be4c-d68f4963b7dd.s3.amazonaws.com/uploads/ckeditor/attachments/7824/3g_Paper_comp osition.pdf).

>>> LENGTH: We estimate the current length of your paper to be ~3400 words, which exceeds our usual limit by a considerable margin. With four display items as at present, the main text of the revised version should be no more than ~1500 words. Keep in mind that important technical details that are not central to the main message of the paper can be moved into the Methods section.

The paper has been shortened as requested. We have used the texcount web service (<https://app.uio.no/ifi/texcount/online.php>) to keep track of the word count in the paper. This service estimates 1506 words for the main text in the amended draft.

>>> SUMMARY PARAGRAPH: All Nature papers begin with a fully referenced paragraph, typically no longer than 200 words (your current abstract is ~285 words). This paragraph starts with a 2- to 3-sentence summary of the "state of the art" immediately prior to this work. It continues with a 1-sentence statement of the main observational result starting with 'Here we report' or an equivalent phrase. The idea is to cleanly separate what you *see* from what you infer from the data. The paragraph concludes with 2 to 3 sentences putting the main findings into the context established earlier so it is clear how the results described in the paper have moved the field forward. In some cases it may be necessary to exceed this limit in order to explain complex material for readers in other fields – in such cases, summary paragraphs can be up to 230 words in length. **The extra length, however, is for introduction and context, and not for additional technical information.** The limit of 230 words is imposed by production, and I have no flexibility. I invite you to iterate with me via email.

>>> What is now the summary section accomplishes some of what we are looking for in a summary paragraph. It too is full of advertising: "the transformative power of JWST", "a new record redshift", "unprecedented detail", "our deepest insight". Just state the facts. If the background context is properly established, readers can judge for themselves what is "powerful" and "record". The section "results" (first paragraph) provides a good background context, but needs to be condensed to a few sentences for use in the summary paragraph.

The previous abstract has been removed and the previous "summary" section has been edited as requested. The authors believe that the information previously included in the first paragraph of "results" (referenced by the editor) is not of key scientific importance (despite being of historical interest), and this has been largely cut from the paper to save space. The text previously in the summary section is more fundamental to the scientific importance of this work. The texcount web service returns a length of 201 words for the new summary paragraph. We would be happy to iterate via email with the editor if this new draft summary paragraph requires further revision.

>>> We are following the AAS journals' practice of *NOT* spelling out JWST at any time -- simply use JWST in all places.

This change has been made.

>>> MAIN TEXT: **Further introductory material in the main text of the paper should not be necessary.**

We have endeavoured to remove introductory material from the main text as requested.

>>> Please remove the subheadings, as they are not necessary in a 1500-word paper. (You can retain them in the Methods section.)

These have been removed as requested.

>>> The section "Spectroscopic data" is very descriptive, so some of that could go to the legend of Figure 1 (legends can be up to 300 words long each), and some to the Methods.

This section has been shortened, with some details moved to the Fig. 1 legend as suggested.

>>> Much of the section "Stellar metallicity" could be omitted, which would address the referees' concerns and also help shorten the paper.

As above, this section has been removed.

>>> Any discussion at the end of the paper should also be brief, and not repeat what is already written in the initial summary paragraph. (Most of the Conclusions section could be omitted.)

This section has been largely cut as requested.

>>> METHODS: At the end of the main text document (after the main figure legends), there should be a section entitled "Methods", which provides a more detailed discussion of the additional methodological information that would allow other researchers to replicate the results (we define "Methods" quite broadly, so this is not limited to details of experimental protocols – supplementary discussion and analysis can also be included). The Methods section will not appear in the print version but will be fully copy-edited and appear online in the full-text HTML and PDF versions. The Methods section should be written as concisely as possible but should contain all elements necessary to allow interpretation and reproduction of the results. If there are additional references in the Methods section, their numbering should continue from the last reference in the main paper, and the list should follow the Methods section. If the methods require chemical structures, figures or tables, these should be supplied as Extended Data (see below). For mathematically complex methods, or methods that require an unusually large number of figures or tables (beyond what can be accommodated as Extended Data), the entire Methods section should instead be supplied as a separate Supplementary Information.

A methods section is included in the draft. Further information previously included in the main text has been moved to this section.

>>> REFERENCES: As a guideline, most papers should need no more than 30 references in the main text; additional references can be cited in (and listed after) the Methods section, as detailed above. Please omit the DOIs. Each reference should include final

page numbers, as well as initial ones, unless the first 'page' is an article number (i.e. A&A beginning in 2010, ApJ beginning in 2011). Our abbreviations for the common astronomy journals are: *Astrophys. J.*, *Astron. J.*, *Astron. Astrophys.*, *Mon. Not. R. Astron. Soc.*. If there are more than five co-authors, please list only the first author's name, followed by et al..

Formatting of the references has been updated to meet these requirements.

>>> MAIN TEXT STATEMENTS: We require authors to provide a detailed Author Contribution statement immediately after the acknowledgements; the specific contributions of each author must be listed. **It is also a condition of publication that authors include an Author Information statement indicating how to access information regarding reprints and permissions, stating whether or not there is a financial or non-financial competing interest, and naming the author to whom correspondence and requests for materials should be addressed.** (This should not be under the affiliations.) Please ensure that this section is included in the manuscript file after the Methods (but before the Extended Data legends) - it will not appear in the print version but will appear online in the full-text HTML and PDF versions. For details of "end note" style and an example see <https://www.nature.com/nature/for-authors/formatting-guide>.

A statement of author contributions is already included in the manuscript. We have additionally included an author information statement as requested.

>>> DATA AVAILABILITY STATEMENT: All published manuscripts reporting original research in Nature Portfolio journals must include a data availability statement. The data availability statement must make the conditions of access to the "minimum dataset" that are necessary to interpret, verify and extend the research in the article, transparent to readers. This minimum dataset may be provided through deposition in public community/discipline-specific repositories, custom proprietary repositories for certain types of datasets, or general repositories like Figshare, Zenodo and Dryad. Providing large datasets in supplementary information is strongly discouraged and the preferred approach is to make data available in repositories. More information on Nature Portfolio's reporting standards and preparing your Data Availability Statement can be found here: <https://www.nature.com/nature-portfolio/editorial-policies/reporting-standards#reporting-requirements>

>>> You cannot say that data are available upon *reasonable* request. The data are either available, or you must *specify* the conditions under which you would decline to provide them.

All data are already publicly available via the Mikulski Archive for Space Telescopes (MAST) with the exception of the spectrum for GS-9209, which was observed as part of JWST programme 2285. These data are proprietary for 12 months from the date of observation (i.e. until 16th November 2023), after which time, they will also automatically

become publicly available via the MAST portal. The data availability statement has been updated to clarify this.

>>> For all studies using custom code or mathematical algorithm that is deemed central to the conclusions, a statement must be included under the heading "Code availability", indicating whether and how the code or algorithm can be accessed, including any restrictions to access. Code availability statements should be provided as a separate section after the data availability statement but before the References. Code should be deposited in a DOI-minting repository such as Zenodo, Gigantum or Code Ocean and cited in the reference list. Authors are encouraged to manage subsequent code versions and to use a license approved by the open source initiative. Additional details can be found here: <https://www.nature.com/nature-research/editorial-policies/reporting-standards#availability-of-computer-code>.

A code availability statement is already included in the manuscript, with all code publicly available via GitHub repositories.

>>> FIGURE LEGENDS: The main figure legends should be listed sequentially after the references in the main text. Each legend should begin with a **brief, non-technical** title that does not refer to panels, features in the figure, or contain any parenthetical information (Fig 3 is not in compliance). Any error bars in the figures must be defined (for example, s.d., s.e.m.) and the value of n indicated; see <https://www.nature.com/nature/for-authors/formatting-guide> for further explanation. I know that in astronomy the default is 1sigma, but different fields use different defaults so we require authors to specify what they use.

The figures and their legends have been moved to the end of the main text references. The title of Fig. 3 has been updated (we presume the use of the technical term "star-formation history" was the concern here?). Where error bars are shown on figures the captions now specify that these are 1 sigma standard deviation values.

>>> FIGURE FORMATTING: Lettering in all figures (labelling of axes and so on) should be in uniform, sans-serif font, in lower-case type, and large enough to permit substantial reduction for publication (minimum font size 5 pt). Separate parts of a figure are labelled a, b, etc **and referred to in that way in the legend**, not "left", "right". Units have a single space between the number and the unit, and follow SI nomenclature or the nomenclature common to a particular field. Thousands are separated by commas (1,000). Unusual units or abbreviations are defined in the legend. Scale bars rather than magnification factors should be used.

The formatting of the figures has been updated as requested.

>>> EXTENDED DATA: Nature is now integrating the supplemental figures and tables into the final version of most papers. Extended Data do not appear in the printed version of the paper but are included online within the full-text HTML and at the end of the

online PDF. Extended Data are an integral part of the paper and only data that directly contribute to the main message should be presented. All Extended Data must be referred to in the main text, figure legends and/or Methods section, and their figure legends should be listed sequentially at the end of the main text, not in the Extended Data files. Authors should assemble the Extended Data into a maximum of ten, A4 size, multi-panelled display items, submitted as individual JPEG, TIFF or EPS files. They must be provided at the same quality as figures for print, but there are important differences in their formatting. More specific instructions are provided in the Extended Data Formatting Guide (http://s3-service-broker-live-19ea8b98-4d41-4cb4-be4c-d68f4963b7dd.s3.amazonaws.com/uploads/ckeditor/attachments/7823/3h_Extended_data.pdf).

>>> Table 1 should be relabeled as Extended Data Table 1, and referred to in that way in the text.

This table has been relabeled as requested.

Response to referees' comments:

Referee #1:

>>> This is a well-written paper on the analysis of a high quality spectrum with JWST of the earliest galaxy with an evolved stellar population. While it is difficult to learn much definitively from a single object, and similar investigations have been published with similar outcomes (e.g., Glazebrook+17, Forrest+20), the galaxy presented here is the most distant yet, and the spectrum is of extraordinary quality. As such its findings represent a leap beyond earlier work, are potentially significant, and in my opinion deserving of consideration by Nature.

>>> While the analysis is quite thorough and I support the overall findings, I have concerns about parts of the analysis that I would like to see addressed, specifically about the star formation history and residual star formation, AGN contribution, and in particular about the inferences on metallicity, which seems to me the weakest part. I also think more information about the methods are needed, as some aspects of the calibration, including absolute flux calibration and resolution, are still in their early days for JWST and may impact the results.

>>> 1) Star formation histories. These are sensitive to the parameterization of the SFH, as I know the authors are well aware of. It raises the question whether the peaked formation time and other aspects of the formation history are driven by the choice of SFH model. At least some different parameterizations should be explored: for example a top-hat with cut-off, or perhaps a "non-parametric" model, which are of course still parametric, but more flexible.

>>> This would certainly imply less precise constraints on the formation and quenching

timescale, and possibly even stellar mass. All of these quantities are reported at percent level accuracy, which are difficult to believe. Likely, the full uncertainty is not captured by sampling the posterior, but hidden in the variety of assumptions adopted in the modeling.

The authors agree with the referee on the importance of this point. We have therefore re-run our fit with both a top-hat model and the continuity non-parametric model of Leja et al. (2019). These fits also use the more permissive prior on the AGN component discussed below. A new section has been added to Methods detailing these tests, with a new Extended Data Figure 2 showing the equivalent of Fig. 3 under the assumption of these two models. We find that our key results are not strongly sensitive to the choice of SFH prior, in particular the formation and quenching times recovered using these models agree to within 1 sigma.

>>> Related, I was wondering to what level obscured star formation can be ruled out, presumably from the ALMA limits. The limit on SFR from H-alpha does not take into account dust. It is argued from the stellar A_v that this should be small, but the star forming component could be obscured differently than the stellar continuum.

>>> What would H-alpha corrected for nebular extinction using the (limit on the) Balmer decrement be? The depth of H-beta may be consistent with a large Balmer decrement. I presume this was not explored because the emission lines and continuum were not jointly fit. Some attempt could be made. The authors can still argue effectively that the high NII/Halpha is better explained by AGN activity or shocks, but it is important to give a full accounting of the possibility of any obscured on-going star formation.

The authors agree on the importance of this point. To address this properly, the authors have expanded the discussion of the SFR of GS-9209, and moved content from several locations in the paper to a new dedicated Methods sub-section, which discusses the ALMA upper limits, as well as the potential effects of dust attenuation on the H alpha line.

>>> 2) Fe/H. This part of the analysis seems curiously weak compared to the rest of the analysis. Without any clear (absorption line) tracer of metallicity the entire analysis depends on constraints from the continuum shape over a rather small wavelength range. This seems tenuous and the implications for metallicity evolution seem not well supported. At the same time, a somewhat higher metallicity would weaken the (already limited) evidence for alpha-enhancement.

>>> Could this be degenerate with many other factors that subtly affect the continuum slope? From absolute flux calibration of the instrument, and polynomial correction, which is not discussed quantitatively in the paper; to the contribution of the AGN continuum, which is subdominant but not negligible; and lastly, perhaps most importantly, the choice of stellar population itself.

>>> If the authors really wish to make a strong case, it may be necessary to explore stellar population models beyond Bagpipes / BC03. Otherwise I think this aspect should be toned down, without further affecting the impact of the paper.

The authors agree with the referees that this is the weakest section of the analysis we present. Given this, and the request from the editor that we shorten the paper, we have removed the discussion of stellar metallicity from the paper.

>>> 3) AGN. I am wondering how well the continuum power law slopes are constrained. I am presuming these parameters were jointly fit, together with the rest of the stellar population model. It struck me that the priors were fairly narrow and that the solutions fell close to the peak of the priors, which suggests to me that it may not be so well constrained by the observations. Are the contributions robust against using a more permissible, e.g., a wider gaussian or even uniform prior? I suspect the continuum contribution could be higher and the shape different. The current reasoning based on the depth of H-beta absorption seems qualitative. Perhaps it does not affect the inferred SFHs much, but it would be good to know. It may lead to more realistic uncertainties.

Yes, the AGN parameters are jointly fit with the rest of the Bagpipes model. We agree that our original choices of Gaussian prior widths on these parameters were potentially narrower than could be well justified. In response to this comment, we have significantly broadened the Gaussian priors on our two AGN power law components, along with extending the range of allowed values to (-2, 2) for both. The Gaussian widths are now 0.5 in both cases, rather than 0.2 and 0.1 for the bluer ($\lambda < 5000\text{\AA}$) and redder ($\lambda > 5000\text{\AA}$) power laws respectively. This has resulted in a redder spectral slope for the bluer power law, and correspondingly higher $f_{5100\text{\AA}}$ flux, which is in better agreement with the Greene et al. (2005) relationship, as predicted by the referee. These changes have made small differences to the inferred SFH, moving the formation and quenching times later by ~ 50 Myr. Where necessary, the values quoted throughout the paper have been updated to reflect this new, more permissive choice of priors.

>>> I am a bit in the dark, to be honest, on how evidence for a modestly over-massive black hole amounts to strong implications that AGN feedback is responsible for quenching the galaxy. I do not understand the argument that this implies causality.

The argument here is that, in order to build up such a large black hole mass, this galaxy must, at some point, have hosted a bright quasar, which would have been more than capable of expelling star-forming gas from the galaxy (see e.g., Maiolino et al. 2012). The authors appreciate however that the likely historical presence of a quasar is not direct evidence for a causal connection between AGN feedback and quenching in this galaxy. We have therefore toned down the language around this connection in the paper.

>>> 4) Methods: More information should be provided on data reduction, calibration, and some aspects of the modeling. These are still early days of JWST and calibration is preliminary. And while the spectra are of impressive quality, it could well be that

systematic errors are dominating the accuracy of the analysis. I suspect that the currently reported uncertainties are optimistic.

>>> Some more information should be provided on the details of the mode, centering of the object in the slit (presumably very good), more detail on the absolute flux calibration of the instrument and proposed uncertainty. Comparison to e.g. NIRCam photometry, although this has its limitations too. A polynomial correction was applied. How big was it. Does this correction affect the measurements significantly? In addition, the pipeline is formally calibrated to a uniformly exposed slitlet, whereas this source is effectively a point source. The resolution of the NIRSpec g235m and g295m is not $R=1000$, but varies by a factor of almost two across the wavelength range. Some of this may work in the authors advantage, in the sense that the resolution is high enough that the results do not depend on detailed treatment, but it would be good to at least discuss. How big was the noise correction?

The requested additional information on the data and observing technique has been added to the first part of the Methods section. Having recalibrated the data using the spectrum of the A-type standard star (which is, of course, also a point source), the spectroscopic data appears to be in very good agreement with the available NIR photometry. Further corrections as a result of our spectrophotometric calibration polynomial are only at the ~ 10 per cent level.

As the referee says, the resolution is high enough that the total dispersion in the data is dominated by the stellar velocity dispersion, rather than instrumental effects, and a comment to this effect has been added to the full spectral fitting part of the methods section. It is also worth noting that the strong Balmer absorption lines, from which the velocity dispersion will be largely constrained, fall towards the centre and the red end of the bluer grating, where the resolution is > 1000 .

The noise correction adds 50 per cent to the pipeline-produced values (which is quite modest compared to similar analyses we have carried out on other datasets). It also appears that this is in good agreement with the underestimation factor found by other groups, which is currently under investigation by the JWST pipeline team (e.g., see <https://github.com/spacetelescope/jwst/issues/7362>). A note on this and link to the GitHub issue has been added to the methods section.

>>> Smaller comments:

>>> 1) The comparison with the sample reported by Labbe+23 is intriguing, albeit speculative until that sample is spectroscopically confirmed. Are these masses all reported on the same IMF?

The authors agree that spectroscopic follow-up of the Labbe+23 objects is of key importance. In the new draft we are now more careful to describe these as "galaxy candidates" throughout the paper. All masses are on the Kroupa IMF.

>>> 2) It would be nice to include measurements on other important lines, specifically [OIII]4960,5008, as this line seems clearly detected, and may provide more clues about the ionization mechanism.

Having experimented along these lines, the authors believe this is inadvisable, as we do not believe robust measurements of [OIII] fluxes can be extracted from our spectroscopic data. Whilst there do appear to be small peaks in emission at wavelengths corresponding to these lines, these are of a similar size to several other local peaks in the continuum level that do not correspond to emission lines.

>>> 3) I am not sure if the IR limits give strong constraints on the amount of obscured AGN activity. I suspect the observational limits are fairly weak, given the high redshift and high expected dust temperature.

The authors have removed the comment to which this refers, and included a discussion of this point in the new Methods sub-section described above.

>>> 4) I presume the difference between the measured sersic index from F125W and this work at F210M is related to the fact that the AGN may contribute more at shorter wavelengths, and that accounting for any contribution in a point source would tend to lead to lower sersic n ? How does not including (or e.g. a 20% contribution) impact the fit?

Yes, this appears to be the case, though even without an AGN contribution we cannot recover the very high sersic index of van der Wel et al. (2014). This galaxy is right at the edge of the size/magnitude constraints they report as being required for a reliable measurement, though it is flagged as reliable. Without the point source contribution, we obtain a radius consistent to within 1 sigma and a slightly increased sersic index of 2.6 (up from 2.3). With a 20 per cent AGN contribution, the sersic index falls to ~ 1.5 and the radius rises to ~ 300 parsecs. In both cases the quality of fit is worse.

>>> I am assuming that accounting for spectral resolution does not play a significant role in the error budget? (see notes on methods). The low dynamical mass is somewhat surprising, although in line with work on dynamical masses of quiescent galaxies at $z=2-4$. If these galaxies end up in the cores (e.g. van Dokkum+17), the expected IMF might be bottom heavy and the discrepancy would become worse compared to the Kroupa IMF assumed here. But I agree that for objects this compact, resolved (lensed) studies are needed until ELTs come online to address this issue.

We have responded to the point about spectral resolution in our response to the referee's major point 4 – yes the total dispersion of the spectrum is strongly dominated by stellar velocity dispersion in the target galaxy. The authors agree with the referee that the IMF is an important systematic uncertainty in this comparison, and a statement to this effect is included in this section of the paper.

>>> 5) A nice overview of the parameters and assumptions is provided. Perhaps a similar overview could be provided for the best fit parameters.

We have added this information as a new Extended Data Table 2.

>>> 6) Some display of the fit residuals would also be welcome.

We have added a new extended data figure showing the fit and residuals for our full spectroscopic and photometric datasets.

Referee #2:

>>> I have read the manuscript "A Massive Quiescent Galaxy at redshift 4.658" by Carnall et al. These authors report new JWST observations of the galaxy GS-9209, which reveal a number of remarkable features. The paper is well-written and the results are both novel and important for our understanding of galaxy formation in the first few hundred Myr of the Universe. I have only one major comment and several minor suggestions which I ask the authors to address.

Major:

>>> 1. I do not believe the authors estimate of the alpha-enhancement of GS-9209. The authors correctly note that both NaD and CaK are very likely affected by absorption by cold gas within the galaxy (whether in the ISM or in an outflow). The fact that NaD absorption is so much stronger than the models argues that NaD is compromised by intervening cold gas, and there is every reason to assume that CaK is similarly affected. It is therefore simply not possible to use those features to estimate $[a/Fe]$. I therefore strongly urge the authors to remove discussion of $[a/Fe]$ from the paper.

The authors agree with the referees that this is the weakest section of the analysis we present. Given this, and the request from the editor that we shorten the paper, we have removed the discussion of stellar metallicity from the paper.

Minor:

>>> 1. The authors use the word "extreme" and similar emphatic words too liberally. I suggest a close edit of the paper to tone down some of the strong wording - the extraordinary nature of this object is manifestly obvious!

These changes have been made.

>>> 2. Do the derived stellar population properties change if both NaD and CaK are

masked in the BagPipes fitting? I'd like to know if the SFH and mass change by more than 1 sigma if these regions are excluded from the fit.

This test was performed during initial preparation of the manuscript – the posteriors do not change noticeably when these two features are masked. As we no longer discuss these features in the manuscript, we feel that including a comment on this probably does not make sense, however we can add this to the draft if the referee would like us to.

>>> 3. Many quantities reported in this paper have error bars that seem unrealistically small (for example the stellar mass with uncertainty of 0.02 dex and BH mass with uncertainty 0.1 dex). I realize the quoted errors are only statistically, and the authors surely appreciate that systematic uncertainties are likely much larger. At minimum, I would like the authors to add a bit of text here and there emphasizing that the uncertainties are stat-only and systematic uncertainties are likely much larger.

A note to this effect has been added to the text of the paper. We also now carry out an investigation of two additional star-formation history models in the Methods section to help quantify the systematic uncertainties introduced by the choice of SFH model.

>>> 4. I would like to see the fit to the photometric SED in the Methods section.

We have added a new Extended Data Figure 1 showing the fit to the photometric SED.

>>> 5. As the authors are likely aware, the empirical MILES library contains very few hot metal-poor stars so one should be careful about using the spectral shape of A stars to estimate metallicity from models based on MILES stars. If the authors have access to theoretical spectral libraries, I would encourage them to try fitting with those libraries to see if a similarly low metallicity is recovered when fitting GS-9209. [This comment is somewhere in between "major" and "minor"…]

As discussed above, we have removed the discussion of metallicity from the paper. The authors will follow up on this issue in future work, and we thank the referee for this suggestion.

Reviewer Reports on the First Revision:

Referee #1 (Remarks to the Author):

I have carefully read the response and updated manuscript and overall I am happy with the new version.

The current version is much improved, more focused and concise, with several weaker aspects removed (e.g. metallicity) and a more toned down language regarding some claims (e.g., AGN feedback). The authors have provided significant additional analysis, and performed all requested tests, inspiring confidence in robustness of the key results.

In particular, I am pleased the authors have explored different parameterizations of the star formation history, which yield consistent results (albeit peaking at somewhat lower SFRs) and seem to support the general conclusions of the paper of a rapid, short star formation period, followed by a long period of quiescence. I am also glad to see that more permissive priors for AGN contribution did not affect the the results.

The more detailed description in methods on observations and data processing, including instrumental effects and noise, and a clear overview of the best-fit parameters, are also very welcome.

There remain only a few points that I would like to comment on.

--- major comments:

1) ALMA

Given the 1-sigma SFR limit of 41 Msun/yr (compared to a quiescent threshold of $SFR \sim 6$) it appears that the ALMA limits are, and will remain, inconclusive, and substantial ongoing dust-obscured SFR with a mass doubling time similar to the inferred stellar age of the object could still occur.

Even a 2-sigma non-detection of a stack of 20 objects, would exceed the adopted quiescent threshold by a factor 3, and would allow forming $>1e10$ Msun over lifetime of the source.

I disagree with the authors that "The extremely blue spectral shape of this object in the rest-frame red-optical to near-infrared (observed-frame 2–8 μm , see Extended Data Figure 1) is also strongly incompatible with a significant obscured star-forming or AGN component."

The simple point I was hoping to make is that the galaxy is exotic (extremely early and dense) and that independent IR constraints on dust re-emission are fairly weak -- and will likely remain so as it will be hard to reach the required depths with ALMA. It is not unthinkable that a nuclear dust-obscured star formation could be optically thick even at observed 8 micron, so it seems prudent to keep that in mind.

But I do not wish to belabor the point further: the ALMA limits are what they are. I would agree that the *simplest* explanation of the SED (spectrum + photometry) is not to infer a highly obscured component. I would encourage the authors to refrain from much stronger statements.

2) Data availability

In this version of the paper the authors have decided to not make available the reduced spectrum, but they now simply refer to future availability of the raw (or basic calibrated) data. I think this is an unwelcome development.

Reducing and calibrating such observations remains highly non-trivial, and may change over time. Access to the reduced spectrum on which these results are based is highly desirable.

I would encourage the authors, in the spirit of enabling reproduction of the results and/or independent analysis, to consider releasing the reduced spectrum and updated photometry in an easily accessible format.

Short of that, I think the authors should at least make available the reduced data upon request.

Minor comments:

1) The authors say that the masses in Fig 3 are all Kroupa IMF, but it appears to me that the points of Labbe+23 sample as plotted in Fig 3. are using an Salpeter IMF.

2) in section "Methods / The star-formation rate of GS-9209", the quiescent sSFR limit is quoted as $\log_{10}(\text{sSFR}/\text{yr}^{-1}) = 0.2/tH$, but this is probably a mix up of linear and log units: $\text{sSFR} = 0.2/tH$

Author Rebuttals to First Revision:

We thank the referee for their further comments, which we have addressed below.

Comments from the referee:

1) ALMA

>>> I disagree with the authors that "The extremely blue spectral shape of this object in the rest-frame red-optical to near-infrared (observed-frame 2–8 μm , see Extended Data Figure 1) is also strongly incompatible with a significant obscured star-forming or AGN component."

>>> The simple point I was hoping to make is that the galaxy is exotic (extremely early and dense) and that independent IR constraints on dust re-emission are fairly weak -- and will likely remain so as it will be hard to reach the required depths with ALMA. It is not unthinkable that a nuclear dust-obscured star formation could be optically thick even at observed 8 micron, so it seems prudent to keep that in mind.

>>> But I do not wish to belabor the point further: the ALMA limits are what they are. I would agree that the *simplest* explanation of the SED (spectrum + photometry) is not to infer a highly obscured component. I would encourage the authors to refrain from much stronger statements.

The authors have toned down this statement - we now state that the blue spectral shape in the rest-frame NIR is consistent with no obscured star-forming or AGN component, rather than being strong evidence against the presence of such a component.

>>> 2) Data availability

>>> I would encourage the authors, in the spirit of enabling reproduction of the results and/or independent analysis, to consider releasing the reduced spectrum and updated photometry in an easily accessible format.

>>> Short of that, I think the authors should at least make available the reduced data upon request.

As suggested by the editor and addressed in our cover letter, we have included in the data availability statement that the reduced data products are available upon request.

>>> Minor comments:

>>> 1) The authors say that the masses in Fig 3 are all Kroupa IMF, but it appears to me that the points of Labbe+23 sample as plotted in Fig 3. are using an Salpeter IMF.

Looking at the now final published version of Labbe+23, this appears to be true, and we have now converted the points shown in Fig. 3 to the Kroupa IMF that is stated in the text.

>>> 2) in section "Methods / The star-formation rate of GS-9209", the quiescent sSFR limit is quoted as $\log_{10}(\text{sSFR}/\text{yr}^{-1}) = 0.2/tH$, but this is probably a mix up of linear and log units: $\text{sSFR} = 0.2/tH$

This was a typo and has been fixed, thanks to the referee for picking up on this and the above point.